# Accurate Measures of Vaccination and Concerns of Vaccine Holdouts from Web Search Logs

Serina Chang[†]
Stanford University
serinac@cs.stanford.edu

Adam Fourney
Microsoft
adam.fourney@microsoft.com

Eric Horvitz
Microsoft
horvitz@microsoft.com

## ABSTRACT

To design effective vaccine policies, policymakers need detailed data about who has been vaccinated, who is holding out, and why. However, existing data in the US are insufficient: reported vaccination rates are often delayed or missing, and surveys of vaccine hesitancy are limited by high-level questions and self-report biases. Here, we show how large-scale search engine logs and machine learning can be leveraged to fill these gaps and provide novel insights about vaccine intentions and behaviors. First, we develop a *vaccine intent classifier* that can accurately detect when a user is seeking the COVID-19 vaccine on search. Our classifier demonstrates strong agreement with CDC vaccination rates, with correlations above 0.86, and estimates vaccine intent rates to the level of ZIP codes in real time, allowing us to pinpoint more granular trends in vaccine seeking across regions, demographics, and time. To investigate vaccine hesitancy, we use our classifier to identify two groups, *vaccine early adopters* and *vaccine holdouts*. We find that holdouts, compared to early adopters matched on covariates, are 69% more likely to click on untrusted news sites. Furthermore, we organize 25,000 vaccine-related URLs into a hierarchical ontology of vaccine concerns, and we find that holdouts are far more concerned about vaccine requirements, vaccine development and approval, and vaccine myths, and even within holdouts, concerns vary significantly across demographic groups. Finally, we explore the temporal dynamics of vaccine concerns and vaccine seeking, and find that key indicators emerge when individuals convert from holding out to preparing to accept the vaccine.

## KEYWORDS

COVID-19, vaccination, search logs, graph machine learning

**ACM Reference Format:**
Serina Chang[†], Adam Fourney, and Eric Horvitz. 2023. Accurate Measures of Vaccination and Concerns of Vaccine Holdouts from Web Search Logs. In *epiDAMIK 2023: 6th epiDAMIK ACM SIGKDD International Workshop on Epidemiology meets Data Mining and Knowledge Discovery, August 7, 2023, Long Beach, CA, USA.* ACM, New York, NY, USA, 19 pages.

## 1 INTRODUCTION

COVID-19 vaccines provide significant protection against severe cases of SARS-CoV-2 [46, 59], yet a large portion of the United

---

[†] Research performed during an internship at Microsoft.

States remains unvaccinated. Effective vaccine policies—for example, where to place vaccine sites [49, 74], how to communicate about the vaccine [18, 72], and how to design campaigns to reach unvaccinated populations [5, 22, 60]—rely on detailed data about who is seeking vaccination, who is holding out, and why. However, existing data are insufficient [43]. Reported vaccination rates are frequently delayed [2], missing at the county-level and below [70], and missing essential demographic data [33, 42]. Surveys provide a starting point for understanding vaccine hesitancy but are often limited by high-level questions [16], small or biased samples [13, 71], and self-reporting biases (e.g., recall or social desirability bias) [3, 66] especially in sensitive contexts such as vaccination [36].

Here, we demonstrate how large-scale search logs from Bing and machine learning (ML) can be leveraged to fill these gaps, enabling fine-grained estimation of vaccine rates and discovering the concerns of vaccine holdouts from their search interests. While search logs are powerful, with widespread coverage, real-time signals, and access to personal interests, the vast amounts of data they provide are unlabeled and unstructured, consisting of billions of natural language queries and clicks on search results. To derive meaning from these queries and clicks, we first impose structure by constructing *query-click graphs*, which encode aggregated query-click patterns as bipartite networks. Second, using a combination of semi-supervised graph ML techniques and manual annotation, we develop two computational resources that enable us to extract vaccine behaviors from large unlabeled search logs.

First, we develop a *vaccine intent classifier* that can accurately detect when a user is seeking the COVID-19 vaccine on search. Our classifier achieves areas under the receiver operating characteristic curve (AUCs) above 0.90 on held-out vaccine intent labels in all states, and demonstrates strong agreement with CDC vaccination rates across states ($r = 0.86$) and over time ($r = 0.89$). Using our classifier, we can estimate vaccine intent rates to the level of ZIP code tabulation areas (ZCTAs), approximately 10x the granularity of counties and preceding lags in reporting. We carefully correct for bias in our estimates from non-uniform Bing coverage, and demonstrate minimal additional bias from our classifier, as it achieves equivalent true and false positive rates across regions.

Second, we construct a novel *ontology of COVID-19 vaccine concerns* on search. Our ontology consists of 25,000 vaccine-related URLs, clicked on by Bing users, that we organize into a hierarchy of vaccine concerns from eight top categories to 36 subcategories to 156 low-level URL clusters. Unlike surveys, our ontology discovers these concerns directly from users' expressed interests and explores them at multiple scales. Furthermore, by measuring individuals' interest in each concern from their clicks, we capture revealed preferences, side-stepping potential biases in self-reporting [24, 66].

Combining our ontology with the vaccine intent classifier allows us to conduct a thorough analysis of how individuals' vaccine concerns relate to whether they decide to seek the vaccine. We use our classifier to identify two groups of users—vaccine early adopters and vaccine holdouts—and compare their search behaviors. We identify significant differences in their vaccine concerns and news consumption; for example, compared to early adopters matched on covariates, vaccine holdouts are 69% more likely to click on untrusted news sites. We find that vaccine concerns also differ significantly even within holdouts, varying across demographic groups. Finally, we analyze the temporal dynamics of vaccine concerns and vaccine seeking, and discover that individuals exhibit telltale shifts in vaccine concerns when they eventually convert from holding out to preparing to accept the vaccine.

Our contributions can be summarized as follows:

(1) A novel vaccine intent classifier, developed with graph ML and human annotation, that achieves AUCs above 0.9 on all states and strong agreement with CDC vaccination rates;

(2) Bias-corrected estimates of vaccine intent rates from our classifier, including estimates for over 20,000 ZCTAs;

(3) A hierarchical ontology of COVID-19 vaccine concerns, including 25,000 URLs clicked on by Bing users, 156 URL clusters, 36 subcategories, and eight top categories;

(4) Analyses of vaccine holdouts' search concerns and news consumption, comparing to early adopters and studying dynamics over time.

We are publicly releasing our code, vaccine estimates, and ontology.[1] We hope that our resources, methods, and analyses can provide researchers and public health agencies with valuable insights about vaccine behaviors, helping to guide more effective, data-driven interventions.

## 2 DATA

Our work uses a variety of datasets, including Bing search logs, CDC vaccination rates, US Census data, and Newsguard labels (Figure 1). Bing is the second largest search engine worldwide and in the US, with a US market share of around 6% on all platforms and around 11% on desktop [65]. Despite having non-uniform coverage across the US, Bing has enough penetration in the US that we can estimate representative samples after applying inverse proportional weighting (Section 4). The Bing data we use consist of individual queries made by users, where for each query, we have information including the text of the query, an anonymized ID of the user, the timestamp, the estimated geolocation (ZIP code, county, and state), and the set of URLs clicked on, if any. Since our work is motivated by insufficient vaccine data and vaccine concerns in the US, we limit our study to search logs in the US market. However, the methods we introduce could be extended to study vaccination rates and vaccine concerns in other languages and countries. We apply our vaccine intent classifier (Section 3) to all Bing search logs in the US from February 1 to August 31, 2021.[2]

[1]https://github.com/microsoft/vaccine_search_study.
[2]February 2021 was the earliest that we could study following data protection guidelines, which allow us to store and analyze search logs up to 18 months in the past. We end in August 2021, since the FDA approved booster shots in September and our method is not designed to disambiguate between vaccine seeking for the primary series versus boosters.

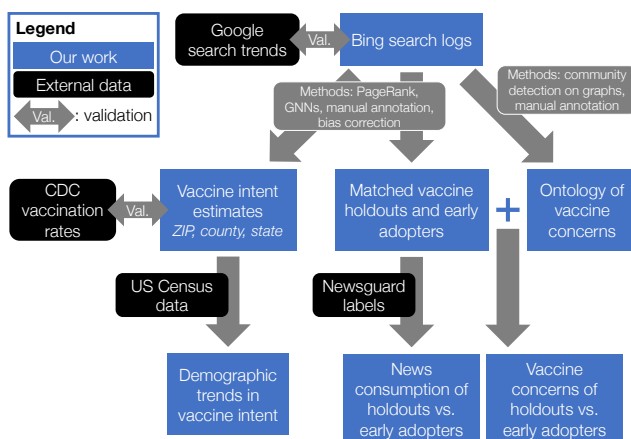

**Figure 1: Our work integrates a variety of datasets and methods to analyze vaccine behaviors from search logs.**

To evaluate our vaccine intent classifier, we compare it to vaccination rates reported by the CDC (Section 4). The CDC provides daily vaccination rates at the levels of states [27] and counties [26]. CDC data are essential but limited, with a substantial portion of county-level data missing. These limitations serve as one of the motivations of our work, since we hope that our vaccine intent classifier can serve as a complementary resource to monitor vaccination rates, especially in smaller regions. To characterize demographic trends in vaccine intent, we use data from the US Census' 2020 5-year American Community Survey [15]. To capture political lean, we use county-level data from the 2020 US presidential election [53]. To quantify the trustworthiness of different news sites, we use labels from Newsguard [52]. Finally, to evaluate the representativeness of Bing search trends, we compare them to Google search trends, which are publicly available online [34].

*Data ethics.* Our work was approved by the Microsoft IRB office and by an internal privacy review process which included officers from both Microsoft Research and the Bing product team. When we use search logs, we are mindful of the need to balance privacy and social benefits when using potentially sensitive user data. While we study individual search logs, since we need to be able to link individual vaccine outcomes (as predicted by our classifier) to search interests, those sessions are assembled using only anonymous user identifiers, which are disassociated from any specific user accounts or user profiles, and cannot be linked to any other Microsoft products. Likewise, in this anonymous view of the logs, location and demographic data were limited to ZIP code-level accuracy. Finally, we are careful to only report results aggregated over thousands of individuals. Aside from Bing search logs, all of the data sources we use are publicly available and aggregated over many individuals.

## 3 VACCINE INTENT CLASSIFIER

Our first goal is to develop a classifier that can accurately detect when a search user is expressing vaccine intent, i.e., trying to get the COVID-19 vaccine (e.g., book an appointment or find a location). Detecting vaccine intent requires precision: for example, if

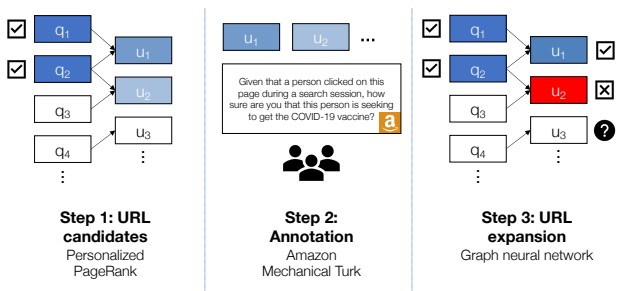

**Figure 2: Our pipeline of methods to identify a large, high-precision set of vaccine intent URLs.**

a user issues the query [covid vaccine], they may be trying to get the vaccine, but they could also be generally curious about vaccine information or eligibility. Thus, we begin by defining a set of regular expressions that allow us to identify vaccine intent queries, i.e., queries that *unambiguously* express vaccine intent. To be included, the query must include both a COVID-19 term ("covid" or "coronavirus") and a vaccine term ("vaccin", "vax", "johnson", etc.). In addition, the query must satisfy at least one of the following criteria: (1) matching some variant of "find me a COVID-19 vaccine", (2) containing appointment-related words or location-seeking words, (3) containing a pharmacy name.

However, in addition to maintaining high precision, we seek to detect as many users as possible who have expressed vaccine intent, so that we have sufficient statistical power for our downstream analyses. Since our search logs contain both queries and clicks, we lose the opportunity to detect many more users if we only detect vaccine intent based on queries. For example, a user may issue the ambiguous query [covid vaccine], but then click on the URL for the CVS COVID-19 vaccine registration page, thus clarifying their intent through their clicks [61]. The challenge with URLs is that they are less formulaic than queries, so we cannot easily define regular expressions to identify URLs expressing vaccine intent.

Our key insight is that, while we cannot use regular expressions to identify URLs, we can use them to identify vaccine intent queries and then use those queries to identify URLs, based on common query-click patterns. For example, vaccine intent queries such as [cvs covid vaccine] or [covid vaccine near me] may result in clicks on the CVS COVID-19 vaccine registration page. To capture these patterns, we construct *query-click graphs* [20, 45], which are bipartite networks between queries and URLs where an edge from a query to a URL indicates how often this query is followed by a click on this URL. Specifically, we construct a query-click graph per US state, aggregating over queries and clicks from two representative months in our study period (April and August 2021). Then, our pipeline proceeds in three steps (Figure 2): first, we use personalized PageRank to propagate labels from queries to URLs, so that we can generate a set of URL candidates (Section 3.1); next, we present the URL candidates to annotators on Amazon Mechanical Turk to label as vaccine intent or not (Section 3.2); finally, we use those labels to train graph neural networks (GNNs) so that we can further expand our set of vaccine intent URLs (Section 3.3).

**Table 1: Top 5 URLs from Personalized PageRank (S-PPR) for the four largest states in the US.**

| State | URL |
|---|---|
| CA | https://myturn.ca.gov/ |
| | https://www.cvs.com/immunizations/covid-19-vaccine |
| | https://www.goodrx.com/covid-19/walgreens |
| | https://www.costco.com/covid-vaccine.html |
| | https://www.walgreens.com/topic/promotion/covid-vaccine.jsp |
| NY | https://covid19vaccine.health.ny.gov/ |
| | https://www.cvs.com/immunizations/covid-19-vaccine |
| | https://www.walgreens.com/topic/promotion/covid-vaccine.jsp |
| | https://vaccinefinder.nyc.gov/ |
| | https://www.goodrx.com/covid-19/walgreens |
| TX | https://www.cvs.com/immunizations/covid-19-vaccine |
| | https://vaccine.heb.com/ |
| | https://www.walgreens.com/topic/promotion/covid-vaccine.jsp |
| | https://corporate.walmart.com/covid-vaccine |
| | https://dshs.texas.gov/covidvaccine/ |
| FL | https://www.publix.com/covid-vaccine |
| | https://www.cvs.com/immunizations/covid-19-vaccine |
| | https://www.walgreens.com/topic/promotion/covid-vaccine.jsp |
| | https://floridahealthcovid19.gov/vaccines/ |
| | https://www.goodrx.com/covid-19/walgreens |

## 3.1 Personalized PageRank for URL candidates

Personalized PageRank [14] is a common technique for seed expansion, where a set of seed nodes in a graph are identified as members of a community, and one wishes to expand from that set to identify more community members [40]. In our case, the vaccine intent queries act as our seed set, and our goal is to spread the influence from the seed set over the rest of the query-click graph. Given a seed set $S$, personalized PageRank derives a score for each node in the graph that represents the probability of landing on that node when running random walks from $S$.

We run personalized PageRank from the seed set of vaccine intent queries (S-PRR) to derive scores for all URLs in each query-click graph. Then, we order the URLs from each state according to their S-PPR ranking and keep the *union* over states of their top 100 URLs as our set of URL candidates, resulting in 2,483 candidates. The number of URLs we have in the union is much lower than the number of states multiplied by 100, since there is overlap between states. However, there is also substantial heterogeneity in top URLs across states, reflecting state-specific vaccine programs and policies (Table 1). By constructing separate graphs and running S-PPR per state, our approach is uniquely able to capture this state-specific heterogeneity. In supplementary experiments, we show that an alternative approach that uses a combined graph over states severely hurts performance for small states (Section A2.2).

S-PPR also provides scores for all queries in the graph, but we found that the seed set was comprehensive in identifying vaccine intent queries. The top-ranked queries that were not in the seed set tended to be location-specific, such as [covid vaccine new york], which is suggestive of vaccine intent but not unambiguous enough. Thus, in the subsequent steps of annotation and GNN expansion, we only seek to add URLs, and consider regular expressions sufficient for identifying queries. However, we also selected a sample

of regular expression-detected queries to present to annotators, to validate whether they were truly vaccine intent. To capture a diverse sample, we use the union over the top 5 and bottom 5 queries per state (ranked by S-PPR), after filtering out queries that were issued by fewer than 50 users, resulting in 227 queries to label.

## 3.2 Annotation on Amazon Mechanical Turk

In this step, we present our URL candidates (and sampled queries) to annotators on AMT. For each URL, we first present it to three annotators. If all three give it a positive label (i.e., Highly Likely or Likely), then we label this URL as vaccine intent. If two give it a positive label and one does not, we assign it to one more annotator, and label it as vaccine intent if that annotator gives a positive label. In other words, we require vaccine intent URLs to receive three positive annotations. With this relatively strict bar, we still find that a large majority (86%) of our URL candidates are labeled as vaccine intent. Furthermore, we observe a clear relationship between S-PPR rank and the percentage labeled as vaccine intent: for example, around 90% of URLs from ranks 0 to 20, around 81% of URLs from ranks 40-60, and around 71% of URLs from ranks 80 to 100 (Figure A2). We also find a very high positive rate (96%) among the queries that we tested, thus validating our regular expressions.

## 3.3 Graph neural networks for expansion

Since manual annotation is expensive, we wish to augment our efforts by training ML models on the AMT labels, then use the models to expand our set of vaccine intent URLs. We formulate this problem as semi-supervised node classification on a graph, since the URLs are nodes in the query-click graph and we are trying to predict whether a URL indicates vaccine intent or not, given labels for a subset of URLs. In this section, we provide an overview of our modeling procedure, with details in Section A1.

*GNN architecture and training.* To solve this problem, we design a GNN [39] that consists of character-level convolutions (CNN) and graph convolutions. We use the CNNs to capture textual information in the queries and URLs, since text can be informative for this problem (e.g., the appearance of "vaccine"). The graph convolutions allow us to learn representations of URLs that draw from the representations of their neighboring queries, which draw from the representations of their neighboring URLs, and so on. In this way, we can capture "similar" URLs in embedding space (similar in terms of both text and graph structure).

To train and test our model, we randomly split the URL labels into a train set (60%), validation set (15%), and test set (25%). However, some states have much smaller graphs, and therefore, fewer positive and negative labels. For example, for Wyoming, we only have 245 positive and 276 negative URLs. We find that with such few labels, the model cannot adequately learn how to predict vaccine intent, with AUCs far below those of large states (Table A1). To address this issue, we *pre-train* the model on S-PPR rankings, which requires no additional supervision. Our intuition is that S-PPR already performed remarkably well at predicting vaccine intent, as we discussed in the prior section. Furthermore, S-PPR rankings do not require any manual labels; we derive them entirely from our initial vaccine intent queries, which were automatically labeled using regular expressions. This pre-training encourages the model

to learn URL representations that are predictive of S-PPR rankings, which we find help substantially with predicting vaccine intent.

*Evaluating GNN performance.* We evaluate model performance by computing its AUC on the held-out test set. Furthermore, to account for randomness from model training and data splitting, we run 10 random trials for every model/state, where in each trial, we re-split the URL labels, retrain the model on the train set, and re-evaluate the model's performance on the test set. First, we find that pre-training significantly improves performance for the smaller states; for example, the mean AUC for Wyoming increases from 0.74 to 0.95 (Figure 3a, Table A1). We find that pre-training seems unnecessary for the larger states, such as Connecticut and Tennesssee, where we are already achieving high AUCs above 0.98. After incorporating pre-training for smaller states (fewer than 5,000,000 nodes), we are able to achieve AUCs above 0.90 for all 50 states and above 0.95 for 45 states (Figure 3b).

*Discovering new vaccine intent URLs.* Finally, we use our trained GNNs to identify new vaccine intent URLs. In order to decide which new URLs to include, we need a score threshold. Our goal is to set the threshold such that any URL that scores above it is very likely to truly be vaccine intent (i.e., we want to maintain high precision). Borrowing the idea of "spies" from positive-unlabeled learning [8], our idea is to use the held-out positive URLs in the test set to determine where to set the threshold. We consider two thresholds: (1) $t_{med}$, the median score of the held-out positive URLs, and (2) $t_{prec}$, the minimum threshold required to achieve precision of at least 0.9 on the held-out test set. Then, we only include URLs that pass both thresholds in at least 6 out of the 10 random trials. Even with this strict threshold, we discover around 11,400 new URLs (Table A2), increasing our number of vaccine intent URLs by 10x. In the following section, we also evaluate the impact of adding these URLs on our ability to estimate regional vaccine intent rates. We find that the new URLs not only increase our coverage of vaccine intent users by 1.5x but also further improve our agreement with reported vaccination rates from the CDC (Table 2).

## 4 ESTIMATING VACCINE INTENT RATES

Using our classifier, we can estimate regional rates of vaccine intent. In this section, we discuss how we correct for bias in our estimates, validate against CDC vaccination rates, and use our estimates to derive insights about fine-grained vaccination trends.

*Bias evaluation.* In Section A2, we decompose potential bias in our approach into two key sources: first, bias from non-uniform Bing coverage, and second, bias from non-uniform true positive rates (TPR) and false positive rates (FPR) of our classifier. We show that, if we can correct for non-uniform Bing coverage and show that our classifier's TPRs and FPRs do not significantly differ across regions, our vaccine intent estimates should, theoretically, form unbiased estimates of true vaccination rates. We evaluate our classifier's TPRs and FPRs on held-out vaccine intent labels, using the same score threshold we used for discovering new vaccine intent URLs. We find that our classifier does indeed achieve statistically equivalent TPRs and FPRs across states (Figure 3b), suggesting that our classifier contributes minimal additional bias. We discuss below how we correct for non-uniform Bing coverage. Additionally, to

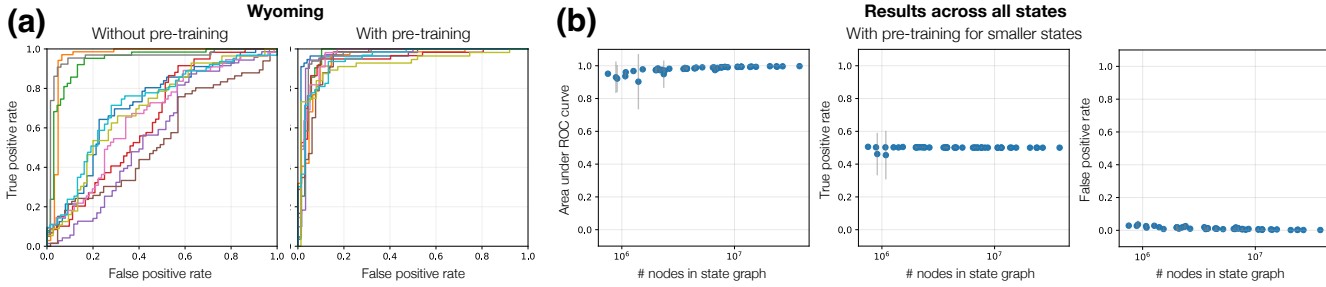

**Figure 3: (a) GNN results with and without pre-training for Wyoming, one of the smallest states. Each line represents one of 10 random trials. (b) Final GNN results for all 50 states, with pre-training for smaller states. Each dot represents a state, with its y-coordinate representing the mean metric over 10 trials and grey bars indicating standard deviation.**

| Pipeline step | CDC corr. | # vaccine intent users |
|---|---|---|
| Only queries | 0.62 | 3.18M |
| +manual URLs | 0.80 | 4.95M |
| +manual and GNN URLs | 0.86 | 7.45M |

**Table 2: Each step of our classification pipeline (Section 3) improves both our correlation with CDC vaccination rates and our coverage of vaccine intent users.**

evaluate the representativeness of Bing data, we compare search trends for vaccine intent queries between Google and Bing and find that, even before applying corrections to Bing data, the trends are highly correlated (Figure A4).

*Estimating coverage-corrected rates.* When we apply our classifier to Bing search logs from Feburary 1 to August 31, 2021, we find 7.45 million "active" Bing users who expressed vaccine intent through their queries or clicks. We focus on active Bing users, i.e., those who issued at least 30 queries in a month, since we can reliably assign them to a location based on their mode ZIP code (or county or state) from those queries. Given a ZCTA $z$, we compute $N(\hat{v}, z)$, the number of active Bing users from $z$ for whom we detect vaccine intent. Furthermore, we estimate the ZCTA's Bing coverage as $\frac{N(b,z)}{N(z)}$, where $N(b, z)$ is its average number of active Bing users over the months in our study period and $N(z)$ is its population size from the 2020 5-year American Community Survey [15]. Then, our coverage-corrected vaccine intent estimate $\tilde{p}(v, z)$ for ZCTA $z$ is

$$\tilde{p}(v,z) = \frac{\frac{N(\hat{v},z)}{N(z)}}{\frac{N(b,z)}{N(z)}} = \frac{N(\hat{v},z)}{N(b,z)}.$$

To estimate the vaccine intent rate for a set $Z$ of ZCTAs, e.g., a state or county, we simply take the population-weighted average.

*Comparison to CDC vaccination data.* When we compare our vaccine intent estimates to state-level vaccination rates from the CDC, we observe strong correlation ($r = 0.86$) on cumulative rates at the end of August 2021 (Figure 4). Notably, we find that the correlation drops to $r = 0.79$ if we do not correct for Bing coverage in our estimates. Furthermore, we find that each step of our classification pipeline—only using queries from regular expressions,

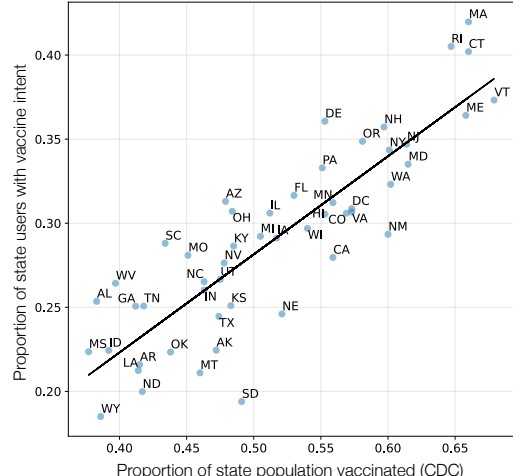

**Figure 4: Comparing CDC state vaccination rates vs. estimated vaccine intent rates from Bing search logs.**

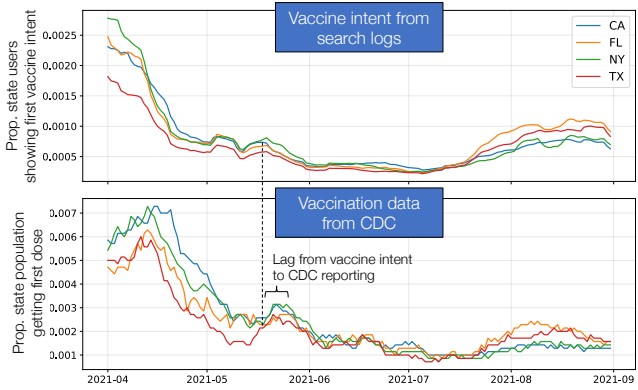

**Figure 5: Rates over time of first vaccine intent (top) vs. first dose from CDC (bottom) for the four largest states in the US.**

incorporating manually annotated URLs from personalized PageRank and AMT, incorporating URLs found by GNNs—improves both our correlation with CDC rates and the number of users we are able

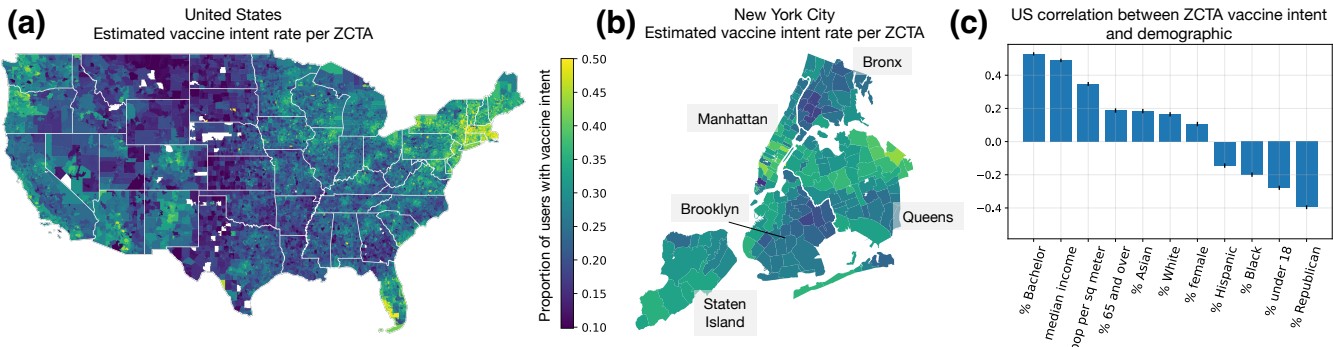

Figure 6: (a) Using our classifier, we can estimate vaccine intent rates per ZCTA, approximately 10x the granularity of counties. (b) Zooming in on New York City shows that estimated vaccine intent rates vary substantially across ZCTAs, even within the same city or county. (c) Correlations between ZCTA vaccine intent rates and demographic variables.

to identify (Table 2). Notably, if we only use queries, the correlation drops to $r = 0.62$ and we lose 57% of the users we identified with our full classifier, demonstrating the value of adding vaccine intent URLs through our graph ML framework.

Additionally, we compare our vaccine intent estimates to the CDC's vaccination rates over time. We observe strong correlations here as well, especially if we allow the CDC time series to lag behind the vaccine intent time series (Figure 5). With lags of 7-15 days (IQR), the median correlation over states reaches $r = 0.89$; without a lag, the median correlation drops to $r = 0.78$. The CDC's lag demonstrates an advantage of our classifier, as it can detect vaccine seeking in real time without delays from reporting.

*Granular trends in vaccine seeking.* Our vaccine intent classifier allows us to pinpoint who was seeking the COVID-19 vaccine, where, and when. We estimate cumulative vaccine intent rates up to the end of August 2021 at the level of ZCTAs (Figure 6a), approximately 10x the granularity of counties, which is the finest-grained vaccination data the CDC provides and, still, with many counties missing or having incomplete data [70]. We observe substantial heterogeneity in vaccine intent at the ZCTA-level, even within the same states and counties. For example, when we focus on New York City, we see that Manhattan and Queens have higher vaccine intent rates, and within Queens, ZCTAs in the northern half have higher rates (Figure 6b), aligning with reported local vaccination rates in New York City [11].

We can also use our estimates to characterize demographic trends in vaccination. When we measure correlations between ZCTA vaccine intent rate and different demographic variables, we find that overall demographic trends from our estimates align closely with prior literature [37, 41, 71, 76]. For example, we observe strong positive correlations with education, income, and population density, and a strong negative correlation with percent Republican (Figure 6c). However, we discover more nuanced trends when we look closer. Demographic trends vary significantly across states (Figure A5), especially for race and ethnicity, and trends change over time. For example, we estimate that older ZCTAs were much likelier to seek the vaccine early in 2021 but this trend fell over time

(Figure A6a), reflecting how the US vaccine rollout initially prioritized seniors [38], and we see an increase in vaccine intent from more Republican ZCTAs in summer 2021 (Figure A6b). Thus, our classifier both confirms existing findings and enables new analyses with finer granularity across regions, demographics, and time.

## 5 SEARCH CONCERNS OF HOLDOUTS

We use our vaccine intent classifier to identify two groups: *vaccine early adopters*, who expressed their first vaccine intent before May 2021, and *vaccine holdouts*, who waited until July 2021 to show their first vaccine intent, despite becoming eligible by April.[3] Comparing the search interests of these two groups allows us to discover relationships between expressed vaccine concerns, news consumption, and vaccine decision-making. To reduce potential confounding, we match each holdout with a unique early adopter from the same county and with a similar average query count, since we know that the populations seeking vaccination changed over time and we do not want our comparisons to be overpowered by regional or demographic differences. In our following analyses, we compare the search interests of the matched sets, with over 200,000 pairs.

*Vaccine holdouts are more likely to consume untrusted news.* First, we analyze the trustworthiness of news sites clicked on by vaccine holdouts versus early adopters. We use ratings from Newsguard, which assigns trust scores to news sites based on criteria such as how often the site publishes false content and how it handles the difference between news and opinion [52]. We find that, in the period while vaccine holdouts were eligible but still holding out (April to June 2021), holdouts were 69% (95% CI, 67%-70%) likelier than their matched early adopters to click on untrusted news, defined by Newsguard as domains with trust scores below 60. Furthermore, we see that as the trust score from Newsguard degrades, the likelier it was that holdouts clicked on the site, relative to early adopters (Figure 7a). For example, sites that are known for spreading COVID-19 misinformation, such as Infowars [25], RT [6], and Mercola [31], were much likelier to be clicked on by holdouts.

---

[3]We did not consider as holdouts those who never showed vaccine intent during our study period, since those users may have gotten their vaccine in ways that are not visible via search data. In comparison, individuals who did not show their first vaccine intent until July 2021 likely did not receive the vaccine before.

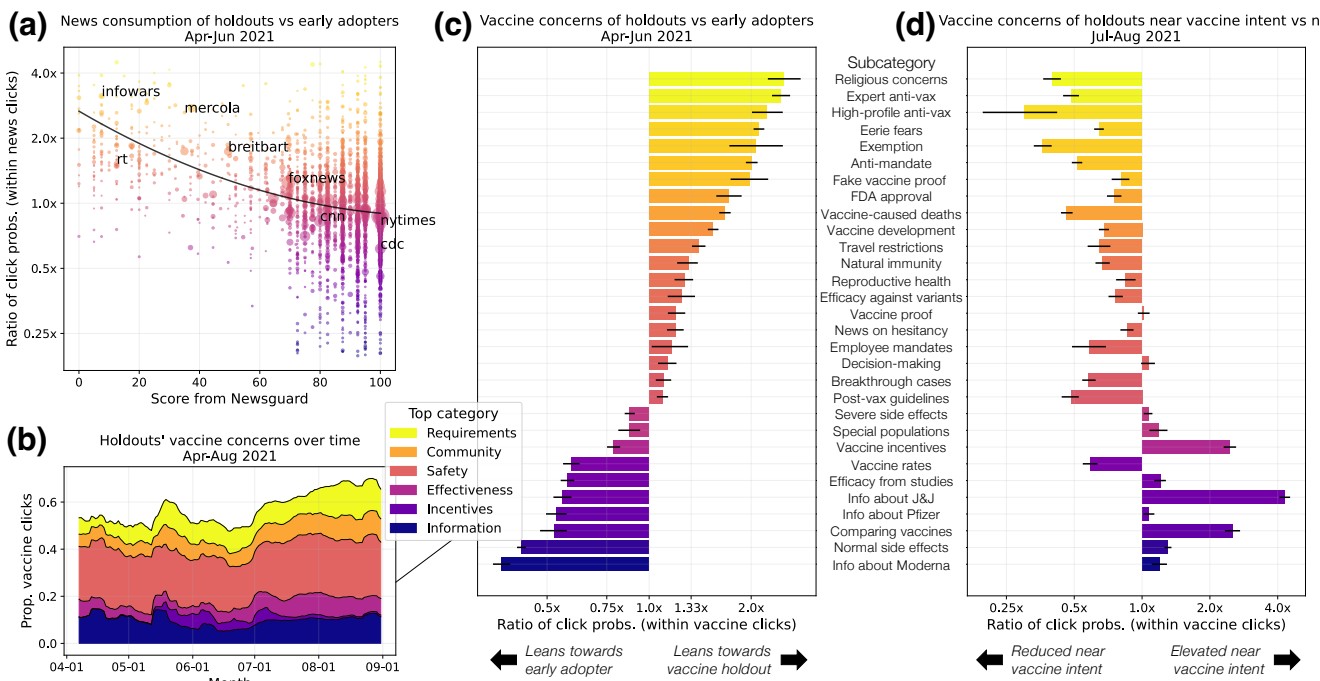

**Figure 7: In all subfigures, news/categories are colored from yellow to dark purple to represent most holdout-leaning to most early adopter-leaning. (a) The lower the trust rating from Newsguard, the likelier it is that vaccine holdouts click on the news site, relative to early adopters. (b) Holdouts' top category concerns include Vaccine Safety, Requirements, and Information, with varying proportions over time. (c) Comparing holdouts vs. early adopters' relative probabilities of clicking on each subcategory (from April to June 2021) reveals each group's distinctive concerns. (d) Near when holdouts express vaccine intent (±3 days) in July and August 2021, their concerns become much more like the concerns of early adopters, with a few important differences.**

*Ontology of vaccine concerns on search.* To characterize vaccine-related search interests in far more detail, we construct a hier-archical ontology of vaccine concerns, defined in terms of 25,000 vaccine-related URLs that were clicked on by early adopters or hold-outs. We construct our ontology from the bottom-up: first, we seek to automatically partition the URLs into clusters. Leveraging graph ML again, we formulate this as a community detection problem on graphs, and apply the Louvain algorithm [12] to the collapsed URL-URL graph (collapsing the bipartite query-click graph over queries). We find that this approach results in remarkably coher-ent clusters (Table A3), due to the strength of the signal contained in query-click graphs, and outperforms standard topic modeling approaches such as LDA [10]. Based on these clusters, we design a comprehensive set of subcategories and top categories, and sort the clusters accordingly. For example, we identify one cluster of news stories announcing vaccine passport requirements in cities, which we sort under the proof of vaccination subcategory and Vac-cine Requirements top category. This bottom-up approach allows us to discover and measure vaccine concerns directly from users' search interests and analyze them at multiple scales, providing complementary insights to more traditional surveys.

In Figure A1, we summarize our resulting ontology, which con-sists of 8 top categories and 36 subcategories. Some top categories encompass a number of distinct subcategories: for example, under

Vaccine Safety, we include normal side effects, severe side effects, concerns about reproductive health, vaccine history and develop-ment, FDA approval, fear of vaccine-caused deaths, and "eerie" fears (e.g., myths about vaccine shedding or becoming magnetic [28]). At the top category-level, we find that vaccine holdouts are, by far, the most concerned about Vaccine Safety, which accounts for 23% of their vaccine-related clicks, followed by Vaccine Information (10%) and Vaccine Requirements (9%). We also observe changes in interests over time (Figure 7b): for example, interest in Vaccine Incentives increased in May 2021, and interest in Vaccine Effective-ness grew in June 2021, following the spread of the Delta variant.

*Distinctive concerns of holdouts vs. early adopters.* Our ontology allows us to compare the vaccine concerns of holdouts and their matched early adopters. First, during the period from April to June 2021, we find that holdouts were 48% less likely than early adopters to click on any vaccine-related URL. Furthermore, their distribution of concerns within their vaccine-related clicks differed significantly (Figure 7c). Using the subcategories from our ontology, we find that holdouts were far more interested in religious concerns about the vaccine; anti-vaccine messages from experts and high-profile figures; avoiding vaccine requirements by seeking exemptions, ban-ning mandates, or obtaining fake proof of vaccination; eerie fears and vaccine-caused deaths; and FDA approval and vaccine develop-ment. In comparison, early adopters were much more concerned

about normal side effects, vaccine efficacy, comparing different types of vaccines, and information about each vaccine (Moderna, Pfizer, and Johnson & Johnson). These differences reveal the importance of a fine-grained ontology; for example, at the top category level, we would see that both groups were interested in Vaccine Safety but miss that early adopters were more concerned about normal and severe side effects, while holdouts were more concerned about eerie fears and vaccine-caused deaths. Our approach also allows us to study *who* is expressing these concerns in greater granularity. Even within holdouts, we observe significant variability in concerns across demographic groups (Figure A7). For example, holdouts from more Democrat-leaning ZCTAs were particularly concerned about FDA approval and vaccine requirements, while holdouts from more Republican-leaning ZCTAs were more concerned about eerie fears and vaccine incentives.

*Holdouts appear like early adopters when seeking the vaccine.* In our final analysis, we exploit the fact that all of our vaccine holdouts eventually expressed vaccine intent to explore how vaccine concerns change as an individual converts from holdout to adopter. From July to August 2021, we analyze how holdouts' vaccine concerns change in the small window (±3 days) surrounding their expressed vaccine intent, compared to their typical concerns outside of that window. We find that in those windows, holdouts' vaccine concerns nearly reverse, such that they look much more like early adopters than their typical selves (Figure 7d nearly reverses 7c). During this time, holdouts become far more interested in the Johnson & Johnson vaccine, comparing different vaccines, and vaccine incentives, and less interested in anti-vaccine messages and vaccine fears. Notably, not all early adopter-leaning concerns reverse as dramatically; for example, even while expressing vaccine intent, holdouts remain less interested in the Pfizer and Moderna vaccines, which may reflect how vaccine hesitant individuals were quicker to accept the one-shot Johnson & Johnson vaccine, instead of the two-shot mRNA vaccines [21, 73]. Furthermore, there are some early adopter-leaning concerns that holdouts do not pick up on during this time, such as interest in vaccine rates. We hypothesize that these concerns are more reflective of an early adopter "persona" rather than of concerns that would become relevant when seeking the vaccine, such as comparing different vaccines.

## 6 RELATED WORK

Our work centers Bing search logs, which have been used to study other health issues such as shifts in needs and disparities in information access during the pandemic [67, 68], health information needs in developing nations [1], experiences around cancer diagnoses [55, 56], concerns rising during pregnancy [29], and medical anxieties associated with online search [75]. Our efforts build on prior work that extracts insights about the COVID-19 vaccine from digital traces, such as social media [50, 57, 58] and aggregated search trends [7, 23, 48]. Our work is also related to other efforts to detect health conditions online, such as predicting depression from social media [19] and monitoring influenza from search queries [32].

Our work seeks to address the challenges of working with digital traces [24, 54] and limitations of prior work [32, 44] by developing ML and human-in-the-loop methods to precisely label search logs

and evaluate bias. Furthermore, as one of the first works to use *individual* search logs to study the COVID-19 vaccine, we have the rare opportunity to link vaccine outcomes (predicted by our classifier) to the same individual's search interests. Our graph ML pipeline is also similar to other "big data" approaches that, due to the scale of unlabeled data, manually annotate a subset of data, train machine learning models to accurately predict those labels, then use those models to label the rest of the data [17, 30, 35, 47]. We extend this approach in several ways, such as by using personalized PageRank to select URLs for more efficient annotation and by setting a strict classification threshold based on "spies" to ensure high precision.

## 7 DISCUSSION

We have demonstrated how large-scale search logs and machine learning can be leveraged for fine-grained, real-time monitoring of vaccine intent rates and identification of individuals' concerns about vaccines. There are limitations to our approach: for example, while we can achieve finer granularity than existing data, we still miss within-ZCTA heterogeneity in vaccine intent. Furthermore, our efforts to minimize bias in our estimates are substantial but imperfect (e.g., we can only approximate TPRs and FPRs of our classifier). We also assume in this work that vaccine intent can be detected through single queries or clicks, but more sophisticated models could incorporate entire search sessions or browsing data beyond search. However, in favor of simplicity and considerations of privacy, we label vaccine intent at the query and click-level.

Despite these limitations, our resources demonstrate strong agreement with existing data and enable analyses that have not been available before. For example, our fine-grained vaccine intent estimates can help public health officials to identify under-vaccinated communities, informing where to place vaccine sites or whom to prioritize in online or real-world outreach programs. Furthermore, our novel ontology and analyses of individuals' vaccine concerns inform how to intervene, guiding messaging strategies for different holdout populations. Lastly, our observation that holdouts resemble early adopters when they eventually seek vaccination indicates that individuals might follow similar paths towards vaccine acceptance. Future work could model these trajectories, try to identify key influences (e.g., vaccine mandates), and use these models to ideally allocate limited resources for interventions.

To facilitate policy impact and future research, we are releasing our vaccine intent estimates and our ontology of vaccine concerns. We hope that these resources will be useful for conducting detailed analyses of COVID-19 vaccine behaviors and vaccination rates. The ontology can also be employed widely in web and social media research; for example, to study how certain classes of URLs (e.g., eerie fears) are disseminated on social media or surfaced by search engines. Finally, we note that our graph ML techniques for intent detection are applicable beyond vaccines, and could be applied to precisely detect other intents of interest, such as seeking stimulus checks or COVID-19 tests. More broadly, we hope that our work can serve as a roadmap for researchers of how to derive rigorous behavioral and health insights from search logs, including how to precisely detect user intents and interests, evaluate and correct for bias, validate against external data, and release resources to promote reproducibility, transparency, and future work.

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

 

# APPENDIX

The Appendix provides additional results and experiments, including detailed descriptions of our ontology (Figure A1), results from developing our vaccine intent classifier (Section A1), our decomposition and evaluations of bias (Section A2), and additional analyses of vaccine intent trends and vaccine concerns (Section A3).

## A1 VACCINE INTENT CLASSIFIER: ADDITIONAL RESULTS

*Annotation results.* As discussed in the main text, in the second step of our classification pipeline, we present URLs to annotators on Amazon Mechanical Turk. We find that a large majority (86%) of our URL candidates are labeled as vaccine intent, when we require at least three positive annotations to qualify a URL as vaccine intent. Furthermore, we observe a clear relationship between S-PPR rank and the percentage labeled as vaccine intent, whether we set the threshold at two or three annotations (Figure A2). For example, when we require three positive annotations, around 90% of URLs from ranks 0 to 20 qualify, around 81% of URLs from ranks 40-60 qualify, and around 71% of URLs from ranks 80 to 100 qualify. Thus, we find that S-PPR predicts vaccine intent remarkably well, with a high rate among its top URLs and agreement with a decreasing rate as the ranking drops.

*Details from GNN experiments.* In the final step of our classification pipeline, we train GNNs to learn vaccine intent labels and discover new URLs. Since there are not enough URL labels from AMT for smaller states, we experiment with *pre-training* the GNN on S-PPR rankings. In practice, before training the model on the URL labels from AMT, we train the model to predict the URLs' S-PPR rankings that we derived in the first step of our pipeline. Since S-PPR rankings become less meaningful in the long tail of URLs, we focus on predicting the top $K = \max(1000, q_{max})$ S-PPR rankings, where $q_{max}$ is the maximum rank (where lower rank corresponds to higher S-PPR score) of the last seed set query.

To test the effect of pre-training on S-PPR rankings, we select six representative states that vary in graph size and US region. We find that pre-training significantly improves performance for the smaller states. For example, the mean AUC for Wyoming increases from 0.74 to 0.95 (Table A1). Specifically, due to the low number of URL labels for smaller states, we observe great variance in the model's performance if we do not pre-train the model, leading to some trials that perform well and some that perform poorly (Figure 3a). Performance becomes far more stable for smaller states after we incorporate the pre-training objective. We find that pre-training seems unnecessary for the larger states, such as Connecticut and Tennessee, where we are already achieving high AUCs above 0.98. So, we set a generous cutoff of 5,000,000 nodes (still larger than the graph size for Connecticut) and we pre-train all states with fewer than 5,000,000 nodes in our data, of which there are 26. After incorporating pre-training for these smaller states, we are able to achieve AUCs above 0.90 for all 50 states and above 0.95 for 45 states (Figure 3b).

As a supplementary analysis, we can also use AUC to evaluate the predictive performance of S-PPR alone and GNN-PPR, i.e., the GNN pre-trained on S-PPR rankings *before* it is also trained on AMT

| State | # nodes | AUC w/o pre-train | AUC w/ pre-train |
|-------|---------|-------------------|------------------|
| WY | 752865 | 0.741 (0.146) | 0.951 (0.014) |
| AK | 909357 | 0.796 (0.187) | 0.921 (0.074) |
| DE | 1269327 | 0.864 (0.134) | 0.968 (0.007) |
| MT | 1533071 | 0.857 (0.139) | 0.978 (0.011) |
| CT | 4407722 | 0.987 (0.005) | 0.984 (0.008) |
| TN | 7712443 | 0.991 (0.003) | 0.990 (0.003) |

Table A1: Effects of pre-training on S-PPR rankings for six selected states. We report the mean and standard deviation of AUC on the test set over 10 random trials.

labels. Here, we evaluate on *all* AMT labels, since none of them were used in constructing S-PPR or GNN-PPR scores. In fact, evaluating on AMT labels is particularly challenging, since we chose to label only the top-ranked URLs according to S-PPR, so we are asking S-PPR to distinguish between URLs that it already considers similar. We conduct this experiment on the 26 smaller states for which we pre-trained our GNNs.

First, we find across these states that S-PPR still performs better than random, with a mean AUC of 0.569, which complements our annotation results showing that even within its top-ranked URLs, S-PPR rankings still correlate with true rates of vaccine intent labels (Figure A2). Second, we find that GNN-PPR consistently *outperforms* S-PPR by 10-15 points, with a mean AUC of 0.675. This is somewhat surprising, since GNN-PPR was only trained to predict S-PPR rankings, without any additional labels. We hypothesize that GNN-PPR outperforms S-PPR because, unlike S-PPR, the GNN can incorporate textual information from URLs and queries, in addition to graph structure. So, while S-PPR incorrectly upweights high-traffic URLs such as facebook.com that are often reached on random walks starting from the vaccine intent queries, GNN-PPR recognizes that these URLs do not look like the rest of high-ranking URLs and correctly excludes them. However, in order to achieve this difference between S-PPR and GNN-PPR, it is important not to overfit on S-PPR. So, we employ early stopping during pre-training; that is, we train the GNN on S-PPR rankings until they achieve a correlation of 0.8 and then we stop pre-training.

Our evaluation results demonstrate that our GNNs are able to accurately predict vaccine intent labels in all 50 states, which is essential as we use our GNNs to discover new vaccine intent URLs. In Table A2, we provide a uniform random sample of the URLs that our GNNs discovered. The majority of them seem to express vaccine intent, with several news stories about new vaccine clinics and information about vaccine appointments. Furthermore, the supplemental analysis of S-PPR and GNN-PPR shows that due to the expressive power of the GNN (with character-level CNN) and the predictive power of S-PPR from a well-designed seed set, we can achieve decent performance without *any* labels at all. These methods, which should be explored more deeply in future work, may be useful in a zero-shot context, allowing lightweight, effective prediction before acquiring any labels.

| Top category | Subcategory | Description |
|---|---|---|
| | Normal side effects | Expected side effects: sore arm, shoulder, fever, etc |
| | Severe side effects | Rare but plausible side effects, severe, potentially long-term: blood clots, myocarditis, etc |
| | Reproductive health | Concerns about fertility, breast feeding, menstruation |
| Safety | Vaccine-caused deaths | Fear of deaths *caused* by COVID vaccine |
| | Eerie fears | Eerie and debunked fears: shedding, magnets, microchips, etc |
| | Vaccine development | History of vaccine development, fear of mRNA technology, ingredients in COVID vaccine |
| | FDA approval | FDA approval of COVID vaccines |
| | Efficacy from studies | How effective the vaccine is, how long immunity lasts, how long for vaccine to take effect |
| Effectiveness | Efficacy against variants | How well does vaccine work against variants (mostly Delta) |
| | Breakthrough cases | Breakthrough COVID cases, symptoms when vaccinated |
| | Natural immunity | Is natural immunity better than vaccine, do I still need vaccine |
| | Vaccine rates | Vaccine trackers, rates of vaccination over time: by state, by country, etc |
| | News on hesitancy | Reporting on vaccine hesitancy and anti-vaxxers, how to talk to vaccine hesitant |
| Community | Expert anti-vax | Anti-vaccine messages from scientists and doctors |
| | High-profile anti-vax | Anti-vaccine messages from high-profile figures: politicians, celebrities, etc |
| | Religious concerns | Religious concerns about the vaccine, seeking advice from religious leaders |
| | Decision-making | Pros and cons of COVID vaccine, should I get the vaccine? |
| | Comparison | Comparing Moderna vs Pfizer vs J&J, side effects, efficacy |
| | Moderna | General news on Moderna vaccine, rollout, side effects, efficacy |
| Information | Pfizer | General news on Pfizer vaccine, rollout, side effects, efficacy |
| | Johnson & Johnson | General news on J&J vaccine, emphasis on blood clots and efficacy |
| | Special populations | COVID-19 vaccine for special populations: autoimmune disease, rheumatoid arthritis, etc |
| | Post-vax guidelines | Guidelines after vaccination: masking, testing, quarantine |
| | Travel | Vaccine requirements to travel: for cruises, other countries, etc |
| | Employment | Employer vaccine mandates: healthcare, government, educators, etc |
| Requirements | Vaccine proof | Required proof of vaccination to enter places: restaurants, gyms, concert venues, etc |
| | Exemption | Seeking exemption on vaccine requirements, religious or medical |
| | Fake vaccine proof | Seeking fake proof of vaccination |
| | Anti-mandate | States banning mandates, lawsuits against employer mandates |
| Incentives | Vaccine incentives | Vaccine incentives: lotteries, gift cards, free groceries, giveaways, etc |
| | Locations | Where to get COVID vaccine (some missed vaccine intent URLs): CVS, Walgreens, etc |
| Availability | Children | Are COVID vaccines for children available / recommended |
| | Boosters | Are boosters available / recommended |
| | New / non-US vaccines | Other COVID vaccines: Novavax, Astrazeneca, Sinovax |
| Other | Non-COVID vaccines | Non-COVID vaccines: flu, MMR, varicella, meningitis, etc |
| | Pet vaccines | Vaccines for pets, mostly dogs and cats |

**Figure A1: Our ontology of vaccine concerns consists of 8 top categories and 36 subcategories.**

| URL | $t_{\mathrm{med}}$ | $t_{\mathrm{prec}}$ |
|---|---|---|
| https://www.chesco.org/4836/61876/COVID-Authorized-Vax | 7 | 10 |
| https://patch.com/new-jersey/princeton/all-information-princeton-area-covid-vaccine-sites | 9 | 10 |
| https://dph.georgia.gov/locations/spalding-county-health-department-covid-vaccine | 9 | 10 |
| https://www.abc12.com/2021/04/22/whitmer-says-covid-19-vaccine-clinics-like-flint-church-are-key-to-meeting-goals/ | 7 | 10 |
| https://www.delta.edu/coronavirus/covid-vaccine.html | 10 | 10 |
| https://www.lewistownsentinel.com/news/local-news/2021/01/scheduling-a-virus-vaccine-appointment/ | 9 | 10 |
| https://www.laconiadailysun.com/news/local/covid-vaccine-clinics-at-lrgh-franklin-now-open-to-public/article_aa4b67e0-601a-11eb-a889-1bd4e6c83de1.html | 6 | 10 |
| https://www.insidenova.com/headlines/inside-woodbridges-new-mass-covid-19-vaccination-site-the-lines-keep-moving/article_eca45b88-8db0-11eb-a649-4bbeccd82cc3.html | 9 | 10 |
| https://www.keloland.com/news/healthbeat/coronavirus/avera-opens-covid-19-vaccine-clinic/ | 10 | 9 |
| https://bangordailynews.com/2021/04/06/news/maine-to-kick-off-statewide-mobile-covid-19-vaccine-clinics-in-oxford-next-week-sk6sr8zcdk/ | 8 | 9 |
| https://morgancounty.in.gov/covid-19-vaccinations/ | 9 | 10 |
| https://www.firsthealth.org/specialties/more-services/covid-19-vaccine | 10 | 10 |
| https://healthonecares.com/covid-19/physician-practices/covid-19-vaccine-information.dot | 9 | 10 |
| https://patch.com/florida/stpete/drive-thru-covid-19-vaccine-sites-open-florida | 9 | 10 |
| https://vaccinate.iowa.gov/eligibility/ | 7 | 10 |
| https://www.baynews9.com/fl/tampa/news/2021/03/17/new-walk-in-vaccine-site-at-tpepin-hospitality-centre-opens-today | 10 | 10 |
| https://www.doh.wa.gov/Emergencies/COVID19/VaccineInformation/FrequentlyAskedQuestions | 10 | 10 |
| https://www.emissourian.com/covid19/vaccine-registration-open-for-franklin-county/article_3638f7a0-5769-11eb-9bba-3f2611173784.html | 10 | 10 |
| https://www.fema.gov/press-release/20210223/maryland-open-covid-19-vaccination-center-waldorf-fema-support | 10 | 10 |
| https://kingcounty.gov/depts/health/covid-19/vaccine/forms.aspx | 10 | 10 |

**Table A2: A random sample (`random_state=0`) of 20 URLs from GNN. $t_{\mathrm{med}}$ and $t_{\mathrm{prec}}$ indicate how often the URL passed the median cutoff and precision cutoff, respectively, out of the 10 trials.**

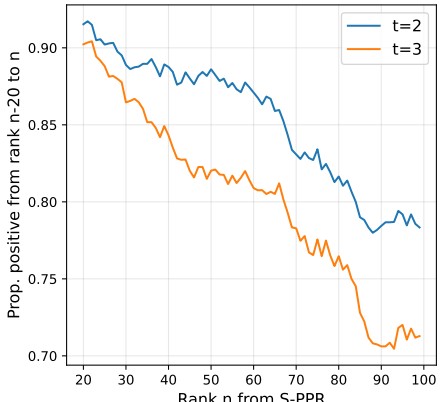

**Figure A2: Comparison of S-PPR rank vs. proportion of URLs around that rank that are labeled as vaccine intent. $t = 3$ and $t = 2$ indicate how many positive annotations were required to qualify for vaccine intent.**

## A2 BIAS DECOMPOSITION AND EVALUATIONS

### A2.1 Decomposition of bias

For a given individual, let $v \in \{0, 1\}$ indicate whether they actually had vaccine intent (up to a certain time) and $\hat{v} \in \{0, 1\}$ indicate whether our classifier labels them as having vaccine intent. Furthermore, let $r$ represent the individual's home region, such as their

state or county. We would like to estimate the regional vaccine intent rate, $\Pr(v|r)$, but we do not have access to $v$, only to $\hat{v}$. To understand how using $\hat{v}$ in place of $v$ may bias our estimates, let us relate $\Pr(\hat{v}|r)$ to $\Pr(v|r)$. First, we introduce another variable $b$, which represents whether the individual is a Bing user. Note that $\hat{v} = 1$ implies that $b = 1$, since our classifier can only identify vaccine intent from users who appear in Bing search logs.

With these variables, we have

$$
\Pr(\hat{v} = 1|r) = \underbrace{\Pr(b = 1|r)}_{\text{Bing coverage of } r} \; [\, \Pr(v = 1|r) \underbrace{\Pr(\hat{v} = 1|b = 1, v = 1, r)}_{\text{Classifier TPR for } r}
$$

$$
+ \Pr(v = 0|r) \underbrace{\Pr(\hat{v} = 1|b = 1, v = 0, r)}_{\text{Classifier FPR for } r} \,].
$$

(1)

$\Pr(b = 1|r)$ represents the probability that an individual from region $r$ is a Bing user, i.e., the Bing coverage of $r$. Incorporating $b, v,$ and $r$ into $\Pr(\hat{v}|b, v, r)$ reflects all of the factors that affect whether the classifier predicts vaccine intent. As discussed, if the user is not a Bing user ($b = 0$), then the probability is 0, so we only consider the $b = 1$ case. If $v = 1$, predicting $\hat{v} = 1$ would be a true positive; if $v = 0$, it would be a false positive. Conditioning $\hat{v}$ on region $r$ reflects the possibility that individuals from different regions may express vaccine intent differently and the classifier may be more prone to true or false positives for different regions. Finally, we make the assumption here that $b \perp v|r$; that is, conditioned on the individual's region, being a Bing user and having vaccine intent are independent. This misses potential within-region heterogeneity,

but to mitigate this in practice, we use ZCTAs as our regions, which are relatively fine-grained.

Based on this decomposition, we can see that if Bing coverage, TPR, and FPR are uniform across regions, then $\Pr(\hat{v}|r)$ will simply be a linear function of $\Pr(v|r)$. Unfortunately, we know that Bing coverage is not uniform. However, we observe $b = 1$ and can assign users to regions, so we can estimate Bing coverage per region and correct by inverse coverage. Thus, our estimate corresponds to a coverage-corrected predicted vaccine intent rate, $\tilde{p}(v,r) = \frac{\Pr(\hat{v}=1|r)}{\Pr(b=1|r)}$. If we refer to the true vaccine intent rate as $p(v,r)$, then we can see that $\tilde{p}(v,r)$ is a linear function of $p(v,r)$ when TPR and FPR are uniform:

$$\frac{\Pr(\hat{v}=1|r)}{\Pr(b=1|r)} = \Pr(v=1|r)\text{TPR} + (1 - \Pr(v=1|r))\text{FPR} \quad (2)$$

$$\tilde{p}(v,r) = \text{FPR} + (\text{TPR} - \text{FPR})p(v,r).$$

Furthermore, if FPR is low, then $\tilde{p}(v,r)$ is approximately proportional to $p(v,r)$. Thus, our first two strategies for addressing bias in our estimates are:

(1) Estimate Bing coverage per region and weight by inverse coverage, which we discussed in Section 4,

(2) Evaluate whether our classifier has similar TPRs and FPRs across regions and whether FPRs are close to 0, which we discuss below.

These efforts are our first two lines of defense against bias. After this, we furthermore compare our results to established data sources, such as the CDC's reported vaccination rates and Google search trends, where we find strong correlations for both.

## A2.2 Evaluating bias in vaccine intent classifier

Our primary source of bias is uneven Bing coverage, which we found can vary by more than 2x across ZCTAs. However, after correcting for Bing coverage, we also want to know that our classifier does not significantly contribute to additional bias. To do this, we must establish that our classifier's TPRs and FPRs do not vary significantly or systematically across regions. The challenge is that we cannot perfectly evaluate these rates, because we do not know all true positives or true negatives. However, we can approximate these metrics based on the labeled URLs that we do have and furthermore make methodological decisions that encourage similar performance across groups.

*Evaluating bias in generating URL candidates.* Recall that in the first step of our pipeline, we generate URL candidates for annotation by propagating labels from vaccine intent queries to unlabeled URLs via personalized PageRank on query-click graphs. Since all URL candidates then go through manual inspection in the second step, we do not have to worry about the false positive rate at this stage. However, we do need to worry about the true positive rate (i.e., recall). For example, if we only kept COVID-19 vaccine registration pages for pharmacies that are predominantly in certain regions, then we could be significantly likelier to detect true vaccine intent for certain states over others. So, through the design and evaluation of our label propagation techniques, we aim to ensure representativeness in vaccine intent across the US.

The most important design decision is that we construct query-click graphs *per state*, then we run S-PPR per graph and take the

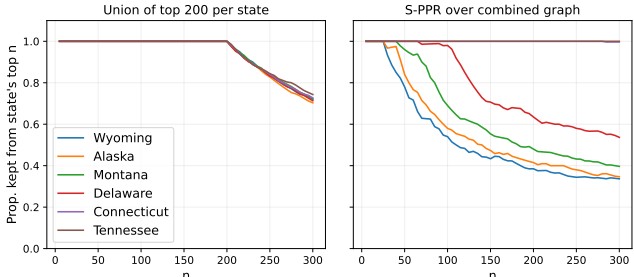

**Figure A3: Comparing our union-over-states (left) to a combined graph approach (right) for generating URL candidates.**

union over states of top URLs as our set of URL candidates. Running this process separately for each state allows us to capture how vaccine intent varies regionally, with state-specific programs and websites for scheduling the vaccine (Table 1). To demonstrate the risks of not using a state-specific approach, we try an alternative approach where we construct a joint graph that combines the queries and clicks for 6 states (the same 6 states as those used in the pre-training experiments of Table A1).

To represent our union approach, we take the union over these 6 states of the top 200 URLs per state, which results in 935 URLs. We compare this to a joint approach, where we take the top 935 URLs from running S-PPR on the joint graph. To evaluate each approach, we compute the proportion of each state's top $N$ URLs that are kept across different values of $N$. While we cannot be sure that every URL in the state's top $N$ is truly vaccine intent, from our annotation results, we saw high positive rates for top-ranking URLs (Figure A2), so we would like to see similar recall at these ranks.

By design, our union-over-states approach ensures equivalent, 100% recall up to $N = 200$ for all states (Figure A3, left). In comparison, we find that the joint approach yields different recalls as early as $N = 30$, with much higher recall for large states than small states (Figure A3, right). For example, it keeps less than 80% of Wyoming's URLs around rank 50 and less than 60% around rank 100, while keeping 100% of Tennessee's throughout. Furthermore, even past $N = 200$, where our union-over-states approach no longer has guarantees, we find that it still achieves far more similar recalls between states than the joint approach. Thus, our design decisions enable similar recalls between states, which helps to reduce downstream model bias. We also cast a wide net when constructing query-click graphs (taking all queries and clicks that co-occur in a session with any query that includes a COVID-19 or vaccine-related word), which may also improve recall and reduce bias, in case our choice of initial keywords was not representative of all vaccine intent searches across the US.

*Evaluating bias in URL expansion from GNN..* In the third step of our pipeline, we use GNNs to expand our set of vaccine intent URLs beyond the manually labeled ones. We would like to see that the performance of GNNs is similarly strong across states, to ensure that the GNN is not creating additional bias when expanding the URL set. We discussed in Section A1 that, after incorporating pre-training on S-PPR rankings for smaller states, GNNs could achieve AUCs above 0.90 for all 50 states. The main metrics of interest

when considering bias, however, are TPRs and FPRs. Unlike AUC, which is evaluated across decision thresholds, TPR and FPR depend on the chosen threshold $t$ above which data points are predicted to be positive. In our setting, we set $t = \max(t_{\mathrm{med}}, t_{\mathrm{prec}})$, since we required new vaccine intent URLs to score above these two thresholds (in at least 6 out of 10 trials): (1) $t_{\mathrm{med}}$, the median score of positive URLs in the test set and (2) $t_{\mathrm{prec}}$, the minimum threshold required to achieve precision of at least 0.9 on the test set. Then, we estimate TPR as the proportion of positive URLs in the test set that score above $t$ and FPR as the proportion of negative URLs in the test set that score above $t$.

We find that TPR is highly similar across states and hovers around 0.5 for all states (Figure 3b, middle). This is because in almost all cases, $t_{\mathrm{med}}$ is the higher of the two thresholds and thus the value of $t$, so the true positive rate lands around 0.5 since $t_{\mathrm{med}}$ is the median score of the true positives. FPR is also highly similar across states and very low (around 0.01; Figure 3b, right), which suggests that the quantity we estimate, $\tilde{p}(v, r)$, is not only a linear function of the true vaccine intent rate, $p(v, r)$, but also approximately proportional to it (Eq. 2). The low FPR is encouraged but not guaranteed by our second threshold, $t_{\mathrm{prec}}$. This threshold ensures that precision is over 0.9, which is equivalent to the false positive rate *among the predicted positives* being below 0.1, which typically corresponds to low false positive rates over all true negatives (which is what FPR measures). The GNN's similar AUCs, TPRs, and FPRs across states, as well as the equivalent recalls in our label propagation stage, increase confidence that our classifier is not adding significant bias to our estimates.

## A2.3 Comparison to Google search trends

Following prior work using Bing data [68], we compare Bing and Google queries to evaluate the representativeness of Bing data.

*Search trends over time.* First, we compare daily search interest in the US over our studied time period from February 1 to August 31, 2021. Google Trends provides normalized search interest over time on Google, such that 100 represents the peak popularity for that time period, 50 means the term is half as popular, and 0 means "there was not enough data for this term." To match this, for a given query, we compute the total number of times it was searched on Bing in the US per day, then we divide by the maximum number and multiply by 100. Again, we apply 1-week smoothing to both the Bing and Google time series. We do not correct the Bing time series with Bing coverage here, since we cannot correct the Google time series with Google coverage, and we want the time series to be constructed as similarly as possible.

We evaluate 30 of the most common vaccine intent queries, including [cvs covid vaccine] and [covid vaccine finder].[4] We observe strong Pearson correlations, with a median correlation of $r = 0.95$ (90% CI, 0.88-0.99) (Figure A4a). These correlations are similar to those reported by Suh et al. [68], who conduct an analogous longitudinal analysis comparing Bing and Google search trends on COVID-related queries and report correlations from $r = 0.86$ to

0.98. Remaining discrepancies between Bing and Google are likely due to differences in the populations using these search engines, as well as potential unreported details on how Google normalizes their search interest trends (e.g., Google may be normalizing differently for [covid vaccine near me], which shows unusual peaks in Google trends and is the the only query for which we do not observe a strong correlation).

*Search trends across states.* Google also provides normalized search interest across US states, where search interest is defined as the fraction of searches from that state that match the query and search interest is normalized across regions such that 100 represents maximum popularity. To imitate this process, we first assign each vaccine intent query to a state based on where the query originated. Then, we approximate the total number of queries (all queries, not just vaccine intent) from each state by summing over the query counts of the active users assigned to each state. We compute the fraction of queries from each state that match the query, then we divide by the maximum fraction and multiply by 100 to normalize across states.

We observe strong Pearson correlations in this analysis too, with a median correlation of $r = 0.95$ (90% CI, 0.57-0.99) across the same 30 vaccine intent queries (Figure A4b). The correlations tend to be stronger on the pharmacy-specific queries, where certain regions dominate, compared to general location-seeking queries such as [covid vaccine near me], which are trickier since they follow less obvious geographical patterns. For the pharmacy-specific queries, we also observe substantial heterogeneity in terms of which region dominates. For example, [publix covid vaccine] is more popular in southern states, with Florida exhibiting the maximum normalized search interest on Google (100), followed by Georgia (26) and South Carolina (20). Meanwhile, [cvs covid vaccine] is more popular in the Northeast, with the top states being Massachusetts (100), New Jersey (96), Rhode Island (90), and Connecticut (65). These differences, reflected in the Bing search trends too, once again highlight the need for regional awareness and representativeness when developing our vaccine intent classifier.

## A3 ADDITIONAL ANALYSES

*State-level demographic trends in vaccine intent.* To investigate more granular demographic trends, we measure correlations per state (only including the ZCTAs in the state) for the 10 largest states in the US. For this finer-grained analysis, we drop percent Republican, since we only have vote share at the county-level, but we keep all other demographic variables, which we have per ZCTA. We find that correlations are mostly consistent in sign across states, but the magnitude differs significantly (Figure A5). For example, the positive correlation with percent 65 and over is around 2x as high in Florida as it is in the second highest states, reflecting the large senior population in Florida and the push for seniors to get vaccinated. In most states, we also see positive correlations for percent Asian and percent White, and negative correlations for percent Black and percent Hispanic, aligning with prior research on racial and ethnic disparities in COVID-19 vaccination rates [51, 63]. Positive and negative correlations for race are particularly strong in certain states, including New York and Florida for percent White/Black, and California and New York for percent Hispanic.

---

[4]We identify 30 representative vaccine intent queries from the top 100 vaccine intent queries, where we choose one standard query for each pharmacy that appears (e.g., [cvs covid vaccine]) and one for each location-seeking query (e.g., [covid vaccine near me]), and drop variants such as [cvs covid vaccines] and [covid 19 vaccine near me].

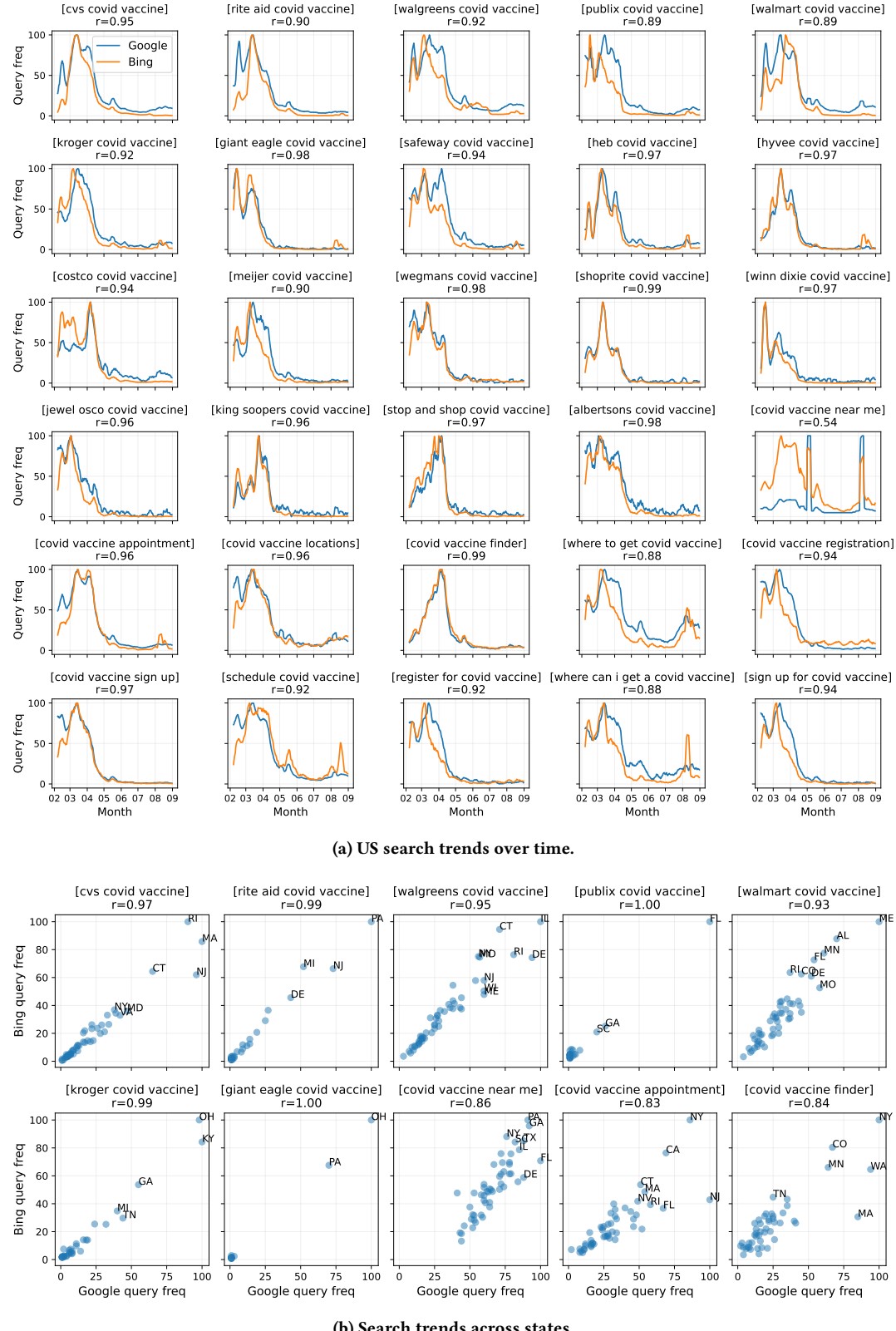

(a) US search trends over time.

(b) Search trends across states.

Figure A4: Comparing search trends on Google vs. Bing for 30 of the most common vaccine intent queries.

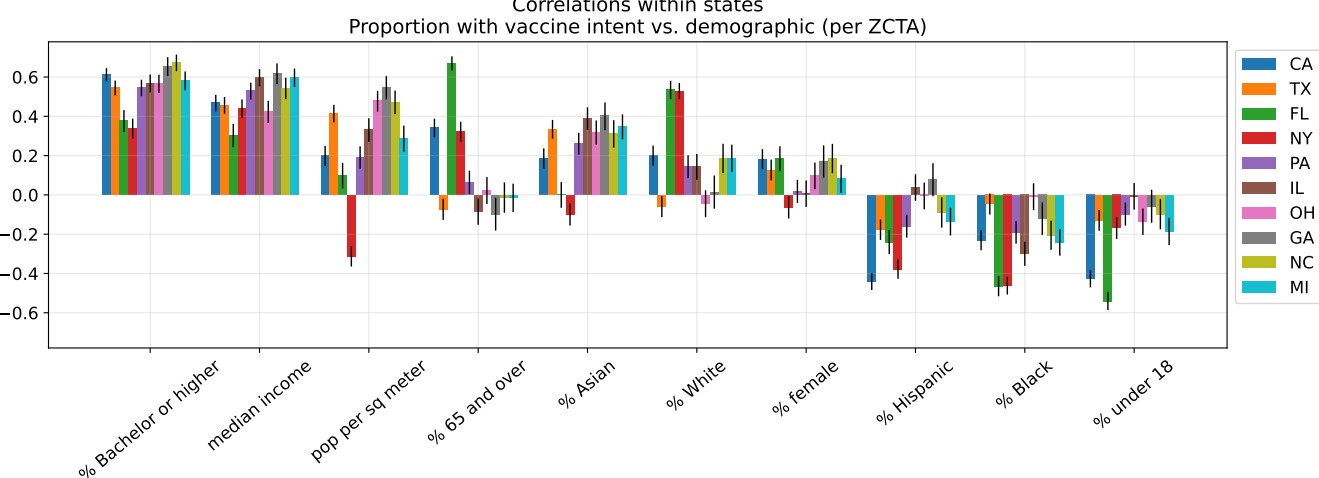

**Figure A5: Correlations between ZCTA vaccine intent rate and demographic variables, for the 10 largest US states. Error bars indicate 95% CIs.**

*Changes in demographic trends over time.* To evaluate changes in demographic trends over time, we separate ZCTAs into top and bottom quartiles, e.g., based on ZCTA median income, and compute each quartile's daily proportion of users showing their *first* vaccine intent. Then, computing the ratio of the top quartile's over bottom quartile's time series reveals changes in demographic trends over time. For example, we estimate that older ZCTAs were much likelier to seek the vaccine early in 2021 but this trend fell over time (Figure A6a), reflecting how the US vaccine rollout first prioritized seniors then expanded to general eligibility [4, 38]. We also see an increase in vaccine intent from more Republican ZCTAs in summer 2021 (Figure A6b), reflecting new calls from Republican leaders to get vaccinated [64] and a self-reported uptick in vaccinations among Republicans [62].

*Examples of URL clusters.* To construct our ontology of vaccine concerns, we begin by automatically partitioning URLs into clusters, using the Louvain community detection algorithm [12] on the collapsed URL-URL graph. We find that our automatic approach produces remarkably coherent clusters, with each cluster covering a distinct topic. The cluster annotations are provided in the ontology that we release, with URLs mapped to 156 unique clusters. We provide a sample of the clusters in Table A3, listing each cluster's most frequently clicked URLs and top query, which we obtain by summing over all queries that led to clicks on URLs in the cluster. From the top query and URLs, we observe distinct topics covered in each cluster: one on CDC masking guidelines after vaccination, one on the Vaccine Adverse Event Reporting System (VAERS), one about religious exemptions for COVID-19 vaccine requirements, and one about side effects of the Johnson & Johnson vaccine.

*Holdout concerns across demographic groups.* We conduct an additional analysis to analyze variation in holdout concerns across demographic groups. For a given demographic variable, we compute its median value across all ZCTAs, split holdouts into those from ZCTAs above the median versus those from ZCTAs below the

median, then compare the vaccine concerns of those two groups of holdouts (by measuring their click ratios). We find significant variability across demographic groups in terms of holdout concerns (Figure A7). Compared to holdouts from more Republican-leaning ZCTAs, holdouts from more Democrat-leaning ZCTAs were far more interested in requirements around employee mandates and vaccine proof, which may be because jurisdictions run by Democrats were likelier to have vaccine requirements [9, 69] while several Republican governors in fact banned such requirements. Meanwhile, holdouts from more Republican-leaning ZCTAs were more interested in eerie vaccine fears, fears of vaccine-caused deaths, and vaccine incentives. We also find that, compared to holdouts from lower-income ZCTAs, holdouts from higher-income ZCTAs were significantly more interested in vaccine requirements, vaccine rates, and anti-vaccine messages from experts and high-profile figures, while holdouts from lower-income ZCTAs were more interested in vaccine incentives and religious concerns about the vaccine.

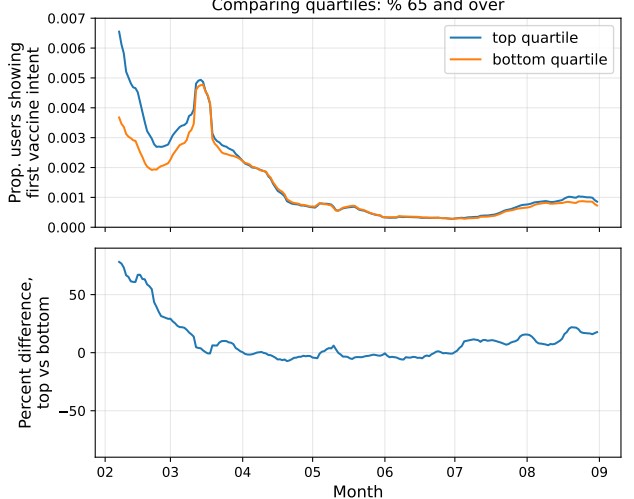

(a) Top and bottom quartiles for percent 65 and over.

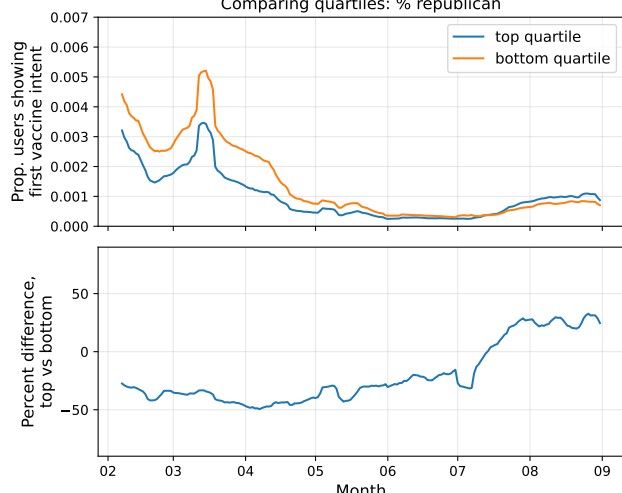

(b) Top and bottom quartiles of percent Republican.

**Figure A6: We quantify changes over time in demographic trends by estimating average vaccine intent rates per quartile over time (top) and computing their percent difference (bottom).**

| # URLs | Top query | Top URLs | % Clicks |
|---|---|---|---|
| 206 | [cdc mask guidelines] | https://www.cbsnews.com/news/cdc-mask-guidelines-covid-vaccine | 8.0 |
| | | https://www.cdc.gov/media/releases/2021/p0308-vaccinated-guidelines.html | 6.9 |
| | | https://www.usatoday.com/story/news/health/2021/05/13/covid-vaccine-cdc-variant-fda-clots-world-health-organization/5066504001 | 4.5 |
| | | https://www.nytimes.com/2021/05/13/us/cdc-mask-guidelines-vaccinated.html | 4.4 |
| 139 | [vaers database covid-19] | https://www.cdc.gov/vaccinesafety/ensuringsafety/monitoring/vaers/index.html | 17.0 |
| | | https://rightsfreedoms.wordpress.com/2021/07/22/vaers-whistleblower-45000-dead-from-covid-19-vaccines-within-3-days-of-vaccination-sparks-lawsuit-against-federal-government | 6.8 |
| | | https://www.theburningplatform.com/2021/07/03/latest-cdc-vaers-data-show-reported-injuries-surpass-400000-following-covid-vaccines | 5.7 |
| | | https://vaersanalysis.info/2021/08/20/vaers-summary-for-covid-19-vaccines-through-8-13-2021 | 4.9 |
| 137 | [religious exemption for covid-19 vaccination] | https://www.verywellfamily.com/religious-exemptions-to-vaccines-2633702 | 16.5 |
| | | https://www.fisherphillips.com/news-insights/religious-objections-to-mandated-covid-19-vaccines-considerations-for-employers.html | 5.1 |
| | | https://www.law360.com/articles/1312230/employers-should-plan-for-vaccine-religious-exemptions | 3.9 |
| | | https://www.kxly.com/who-qualifies-for-a-religious-exemption-from-the-covid-19-vaccine | 3.3 |
| 113 | [johnson and johnson side effects] | https://www.openaccessgovernment.org/side-effects-johnson-johnson-vaccine/109505 | 20.3 |
| | | https://www.healthline.com/health/vaccinations/immunization-complications | 8.1 |
| | | https://www.msn.com/en-us/health/medical/these-are-the-side-effects-from-the-johnson-and-johnson-covid-19-vaccine/ar-bb1f03fq | 4.3 |
| | | https://www.healthline.com/health-news/mild-vs-severe-side-effects-from-the-johnson-and-johnson-covid-19-vaccine-what-to-know | 4.3 |

**Table A3: The 4 highest-modularity clusters with at least 100 URLs. For each cluster, we provide its number of URLs, its most frequent query, its top 4 URLs (by click frequency), and percentage of clicks over all clicks on URLs in the cluster that the URL accounts for.**

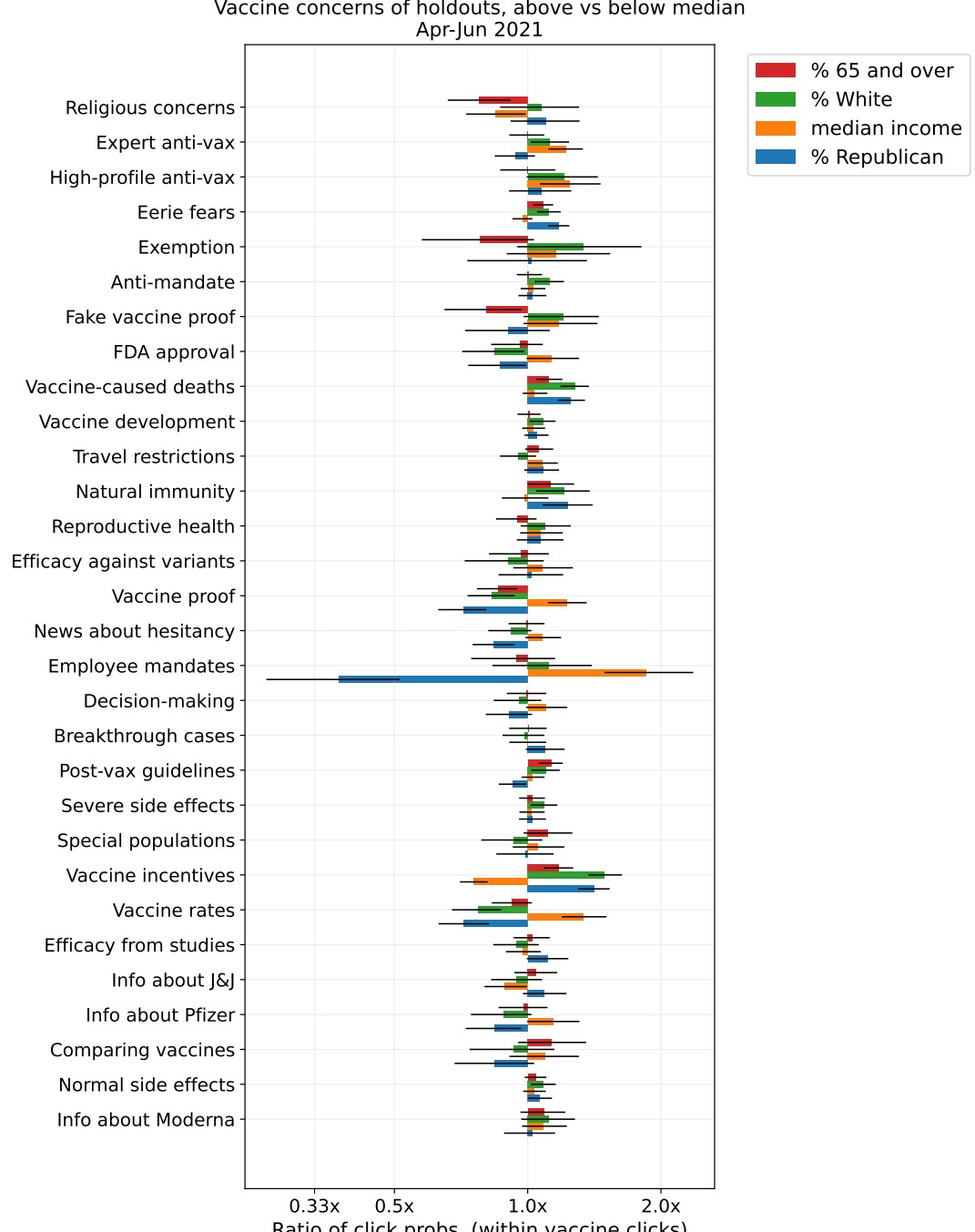

**Figure A7: Variability in holdout concerns across demographic groups. For each demographic variable (e.g., percent Republican), we compare the concerns of holdouts from ZCTAs above the variable's median versus holdouts from ZCTAs below the median. Subcategories are ordered from most holdout-leaning to most early adopter-leaning, following Figure 7c. Error bars indicate bootstrapped 95% CIs.**

