# OpenReview forum: "Accurate Measures of Vaccination and Concerns of Vaccine Holdouts from Web Search Logs"
_KDD.org/2023/Workshop/epiDAMIK — KDD 2023 Workshop epiDAMIK_

### Official Review · Reviewer_iq46 · 2023-06-16
**Very well-motivated problem; well-designed computational study of health policy**

**Rating:** 5
**Confidence:** 4

**Review:**

The paper proposes and implements a framework for fine-grained estimation of vaccination rates across geographical locations, vaccine holdouts, and the behavior of vaccine holdouts over time. The authors leverage a combination of search engine query data, aggregate vaccination rates, census data, and news reliability ratings (i.e., Newsguard) for their method. This is a particularly challenging problem due lags in vaccination reporting and self-reporting biases, especially among holdouts. The authors demonstrate that their vaccine intent classifier performs well and correlates with CDC vaccination rates, and conduct a fine-grained analysis of concerns among vaccine holdouts over time.

The real-world impact and applicability of this paper is obvious to me. The authors select a very topical and compelling area (COVID-19 vaccination hesitancy) as well. Although my experience is primarily computational, the results seem grounded in vaccine policymaking objectives/priorities as well. This work further provides a template that could be potentially adapted to other policy rollouts both retrospectively (e.g., ACA rollout) and the future, provided that the requisite data sources are available.

The comparison of query data between different sources (Bing vs. Google) also addressed my biggest concern — i.e., how representative is the population studied. I also found the breakdowns of vaccine intent by demographic to be very compelling (Fig. 6c, and A5).

A few questions about the method:
* Since there are so many steps, the pipeline for generating vaccine intent labels seems susceptible to error propagation (i.e., if there is a systematic bias in the human annotators, or earlier in the pipeline) since it depends on the quality of data collected— what checks, in addition to those mentioned in the paper (some human evaluation & comparison of Google vs. Bing query data), were done for systematic biases/other pitfalls at each stage of the pipeline?
* It is slightly unclear to me how negative vaccine intent examples were labeled. Is this based on the human annotation method in Sec. 3.2 (i.e., <3 positive annotations), followed by GNN-based label-propagation + spies? What if we label vaccine intent using a simple majority vote method (i.e., 2-1 is sufficient) at the human annotator phase? Are queries that have nothing to do with COVID-19 or vaccinations ever included as negative examples?

 Some further questions about the results:
* In Fig. 6a, some counties are shown in white. Is this because the sample size is too small to generate an estimate of vaccine intent?
The authors choose Newsguard as a provider of news reliability ratings; however, such ratings are inherently dependent on the rating provider’s specific methodology (i.e., who decides who is more reliable in an increasingly polarized news environment). Are there alternate providers of trust ratings, and are the results robust to such changes?
* How were the URL clusters validated? How was model selection (i.e., Louvain over LDA) performed? What is the definition of a “remarkably coherent cluster?” While all of the results look believable, I would have liked to see some measurement of cluster quality here (although this is difficult to do objectively) in addition to the qualitative analysis. Or, is there a human-annotator based way to partially validate these clusters?
* I don’t know that “Holdouts appear like early adopters” is the correct framing towards the end of Sec. 5 — I would expect 7d to look much flatter (vertically) if that were the case, which is true for a few of the bars, but instead I mostly notice the reversal. So it seems like the correct conclusion is that some holdouts’ concerns dramatically shift w.r.t. early adopters at some point, while others converge towards early adopters’ concerns. The reversal trend is probably the most interesting piece in my opinion.

Additional breakdowns of the results that I would find interesting:
* Stratification by area deprivation index, tribal vs. non-tribal, rural vs. urban (Pop/sq. m. is a proxy), access to healthcare (e.g., # of pharmacies offering the vaccine per capita/within 1h)

I also wanted to raise a potential ethical consideration for future work — due to the cross-platform aggregation of data required, the potential for privacy violations due to invasive behavioral interventions or discrimination should be considered in my opinion — for example, targeting specific users for misinformation, vaccine providers/pharmacies engaging in implicit adverse selection by targeting specific segments, or discriminatory labor practices based on vaccine status. One could replace the word “vaccine” with “health” for similar studies on health policy as well.

Since this study largely consists of retrospective data analysis, the risk to users’ privacy is very small at this stage. While I think the authors exercised due diligence in data ethics via IRB approval, anonymization, dissociation from specific user accounts/profiles, ZIP-level granularity, and ensuring no linkage to other products is possible, I am wondering about the potential for actors that do not exercise the same standards of diligence as the authors to harm users’ privacy.  I.e., could a bad actor copy this code and engage in behavioral interventions/discriminatory practices, and what safeguards, computational, legal, or otherwise, exist to mitigate any such threats?

Overall, I think the authors did develop a rigorous and well-motivated method for classifying vaccine intent via a multi-stage pipeline featuring regex queries, URL identification via a combination of PPR, human annotation, a GNN, and the Spy technique from PU learning. The fine-grained analysis of the model's predictions then provide insights into vaccine hesitancy rates, and how concerns of vaccine holdouts change over time. I find that this is already a well-motivated, clear, and well-written computational study of vaccination policy, and addressing the above would simply strengthen the work further in my opinion.

---

### Official Review · Reviewer_ZmWp · 2023-06-19
**The authors did an in-depth analysis of the search logs related to the vaccines to detect an individual’s vaccine intent and further discovered insights on the behavioral difference of (i) early vaccine adaptors and (ii) vaccine-resistant groups. Overall, the paper is well written, is original, and would help the community to understand behavioral patterns from the web search logs.**

**Rating:** 5
**Confidence:** 5

**Review:**

Summary (Long)
- The authors did an in-depth analysis of the search logs related to the vaccines to detect an individual’s vaccine intent and further discovered insights on the behavioral difference of (i) early vaccine adaptors and (ii) vaccine-resistant groups. Their pipeline of the vaccine intent classifier includes finding top candidates for user URLs, using personalized PageRank, followed by annotation via crowdsourcing, and expanding URLs via GNNs. They also prepared an ontology of vaccine concerns by applying a community detection algorithm. Though some of the decisions of model choices are not well justified, overall, the paper is well written, is original, and would help the community to understand behavioral patterns from the web search logs.

Strong points (Pros)
- Overall, their method could fill the gaps in understanding individual vaccine intentions and behaviors through web search logs.
- Their vaccine intention classifier design is well-motivated, easy to follow, and performs well at 0.9 AUC.
- Authors did an in-depth study on this problem and provided enough details and additional analyses in the appendix.

Weak points (Cons)
- The evaluation of their vaccine intention classifier is insufficient, especially because their model is not compared with other baseline methods. If there are no direct methods to evaluate, the authors should do some literature review on somewhat relevant papers that uses search logs in predictive modeling and have those as a set of baselines to compare the performance of the method.
- Design decisions of their modeling are often not justified. E.g., in section 3.1, the authors chose to use personalized page rank as it is a common technique for seed expansion methods. In fact, seed set expansion itself is a well-studied problem, and there exist many more methods developed for this problem in the past decade. I’d suggest authors review state-of-the-art methods in the seed set expansion problem and explore some other methods in their pipeline. Some examples are:
    - Whang, Joyce Jiyoung, David F. Gleich, and Inderjit S. Dhillon. "Overlapping community detection using seed set expansion." Proceedings of the 22nd ACM international conference on Information & Knowledge Management. 2013.
    - Li Y, He K, Bindel D, Hopcroft JE. Uncovering the small community structure in large networks: A local spectral approach. In Proceedings of the 24th international conference on world wide web 2015 May 18 (pp. 658-668).
- Authors claim that vaccine concerns differ significantly within holdouts. If this is true, I am worried that the performance of the ‘binary classifier,’ the vaccine intention classifier, may be suboptimal because there could be a large variance in those in holdouts. In such cases treating the problem as clustering and finding the clusters of holdouts with similar vaccine concerns may make more sense.

Minor comments
- In the abstract, please provide some details about your claims. E.g. the first claim is ‘vaccine intent classifier that can accurately detect …’ – here, please provide how accurate it was. Also, in the abstract ‘… find that key indicators emerge…’ – please list the indicators (maybe provide the most important ones).
- The captions for the tables should be placed on the top of the table, not below the table.
- Please justify the usage of CNNs for capturing textual information in the queries and URLs.
- Please justify using the Louvain algorithm for the community detection problem in section 5.
- There's a typo in section 3.1 . Please change S-PRR to S-PPR

---

### Official Review · Reviewer_pntf · 2023-06-30
**Measuring vaccine intent using web search data**

**Rating:** 4
**Confidence:** 3

**Review:**

The main contribution of this paper is a COVID-19 vaccine-intent classifier that can potentially give an accurate measure of vaccine hesitancy in an individual by analyzing the search history. This classifier is trained on search queries and website clicks from Bing search logs and annotated using Amazon Mechanical Turk.

Another contribution is an ontology of website URLs which consists of 25,000 vaccine-related URLs, organized into eight top categories to 36 subcategories to 156 low-level URL clusters. They combine this ontology with their vaccine-intent classifier and got improved performance.

The classifier correlates with the CDC vaccination data in the sense that states having high vaccination rates have low vaccine-hesitancy and states having low vaccination rates have high vaccine-hesitancy.

One weakness is that they cap their analysis till August 2021, since the FDA approved booster shots in September and their method cannot distinguish between vaccine seeking for the primary series versus boosters. But it still would have been interesting to see how the classifier performs beyond August 2021. Also it is not clear how this method will perform with other vaccines that are not as popular as the COVID-19 vaccine.

But overall the contribution is nice and I think it should be accepted.

---

### Official Review · Reviewer_EV5H · 2023-06-30
**Detecting vaccine intent from user search behavior**

**Rating:** 5
**Confidence:** 4

**Review:**

The authors study the problem of detecting vaccine intent from Bing search query log data. Briefly (as I understand their method) their goal is to take a query + click graph and label it with whether it represents vaccine intent or not and then use the results of this classification to estimate the number of vaccines that will be administered in a particular zip code tabulation area. To do so, the authors use Mechanical Turk to label an initial set of query-URL click pairs and then apply semi-supervised learning techniques to grow this set of labels. Pretraining in the form of initializing the model to minimize an auxiliary loss is applied to states with less data. The resulting classifier is evaluated to be highly effective at detecting vaccine intent. Then, a bias correction is performed to go from Bing user counts to population counts, as the usage of Bing is not uniform across states. The estimates the authors develop are highly correlated with CDC-reported vaccine counts, but more granular and do not have a reporting delay.

The paper is of high quality, generally clear, makes methodological innovations, and likely to be of wide interest.

Minor comments:
- Section 3 para 1—fairly important to include the precise criteria for inclusion (at least in Appendix)
- Giving some overview of the challenge of detecting intent from queries would be helpful for those who have not worked with this kind of data before. For example, in 3.1, the phrase “covid vaccine New York” is mentioned as suggestive but not unambiguous enough. But it is not clear what is missing from this. Is it that the location named is not specific enough? Or is covid vaccine + location always too ambiguous?
- How were URLs presented to the annotators? Did they see just the URL or did they see the page it led to?

Things that came to mind:
- Accuracy of intent classification across time—I believe this is not reported anywhere. This is a pretty important question given the Google Flu Trends experience.
- Connect vaccine intent queries to queries about symptoms, e.g., does experiencing symptoms motivate people to seek vaccine information?